# Chemical Variability and Chemotype Concept of Essential Oils from Algerian Wild Plants

**DOI:** 10.3390/molecules28114439

**Published:** 2023-05-30

**Authors:** Fatima Zahra Benomari, Mathieu Sarazin, Djamel Chaib, André Pichette, Hinane Boumghar, Yacine Boumghar, Nassim Djabou

**Affiliations:** 1Centre Universitaire de Maghnia, Route de Zouia, N99, Maghnia, Algeria; f_benomari@yahoo.com; 2Laboratoire COSNA, Faculté des Sciences, BP 119, Université de Tlemcen, 13000 Tlemcen, Algeria; 3CÉPROCQ, Collège de Maisonneuve, 6220 Rue Sherbrooke Est, Montréal, QC H1N1C1, Canada; msarrazin@cmaisonneuve.qc.ca (M.S.); hboumghar@cmaisonneuve.qc.ca (H.B.); 4BIO-SOURCE S.A.R.L., 28 Ferme Kheloufi, Zeralda 16000, Algeria; djamel.chaib@biosource-dz.com; 5Centre de Recherche sur la Boréalie (CREB), Laboratoire LASEVE, Université du Québec à Chicoutimi (UQAC), 555 Boulevard de l’Université, Chicoutimi, QC G7H2B1, Canada; andre_pichette@uqac.ca

**Keywords:** essential oils, chemical variability, medicinal plants, chemotype, applications

## Abstract

The chemical compositions of eleven wild species of aromatic and medicinal plants indigenous to Algeria, including Thymus, Mentha, Rosmarinus, Lavandula, and Eucalyptus, were analyzed. The identification of the chemical composition of each oil was conducted using GC-FID and GC-MS capillary gas chromatography. The study investigated the chemical variability of the essential oils based on several parameters. These included the impact of the vegetative cycle on oil composition, variations among subspecies of the same species, variations among species within the same genus, the influence of environmental factors on composition variations within a species, chemo typing, and the genetic factors (such as hybridization) contributing to chemical variability. The concepts of chemotaxonomy, chemotype, and chemical markers were examined to understand their limitations and emphasize the importance of regulating the use of essential oils derived from wild plants. The study advocates for an approach that involves the domestication of wild plants and screening their chemical compositions according to more specific standards for each commercially available oil. Lastly, the nutritional implications and the variability of nutritional impact based on the chemical composition of the essential oils will be discussed.

## 1. Introduction

Modern lifestyles expose humans to multiple sources of stress and pollution, industrial foods being on first the line, especially since they have been proven to have risks for dysregulating the immune system, as well as for disruptions of the endocrine system [1]. As an alternative, natural products based on aromatic and medicinal plants and their extracts, such as essential oils, have gained interest over the last decade.

The use of plants and, in part, essential oils to neutralize pathogens and restore immunological balance dates to ancient civilizations [2]. Today, technology allows us to characterize these plants, and analytical techniques identify chemical composition of their extract [3]. Hence, the research on essential oils comes out of a scientific curiosity but also to initiate healthy lifestyles to use natural products, whose risks are still under investigation [4]. Several industries such cosmetics, natural health products, etc., include plant extracts in their products. They have become available ingredients in pharmacies, organic shops, and supermarkets, which are governed by different quality control regulations [5].

Essential oils are fragrances to use for cosmetics but also for therapeutics due to their acidic forms. They regulate organic alkalinization to avoid infections. They help to control stress, promote the conservation of energy and tone, and slow down cellular ageing [2]. Their antimicrobial activity increases the shelf life of foods [6]. Some essential oils are antifungal agents useful for food preservation [7,8,9,10,11,12,13,14,15,16]. In addition, these volatile molecules are used in agriculture as soil correctors (redox couple) due to their targeted action and biodegradable nature [17,18].

However, the bioactivity depends on the chemical profile of the plant [4], thus careful analyses of the risks and targets should be conducted before use. For example, the composition of the essential oil of two plants of the same botanical species is not constant. Under the influence of external factors, it may even present different biochemical specificities [19].

Some studies have shown that the presence of different chemotypes modifies the activity of the oil and its bioactivity [20,21]. The chemical composition of an essential oil varies with hybridization, environmental factors, and cultural practices. For example, the essential oil of peppermint (*Mentha x piperita* L.) is rich in (-)-menthol, the enantiomer responsible for the well-known minty fragrance. However, when grown with different fertilizers, it is replaced by (+)-menthol, whose scent is considered unpleasant [22]. Similarly, the addition of nitrogen has been shown to increase biomass yield and delay flowering development, resulting in a higher yield of essential oil but with less menthol and more menthone. In contrast, the addition of potassium forces the plant to mature and decreases the yield of essential oil, which contains more menthol and methyl acetate [23]. This means that individuals of the same botanical species, with the same genome and phenotype, may differ in chemical composition and thus biological activities. Factors affecting the plant’s composition such as time of harvest, vegetative cycle, soil quality, environmental factors, climate, altitude, and hygrometry are also to be taken into consideration [24,25,26].

This study is part of a research work conducted to understand and use, in the right way, essential oils in Algeria [27,28,29,30,31,32,33,34,35,36,37,38,39,40,41,42,43,44,45], with a specific objective to identify the factors that influence the chemical variability of the essential oils from wild Algerian plants of the genera *Thymus*, *Mentha*, *Rosmarinus*, *Lavandula*, and *Eucalyptus* in collaboration with an industrial partner (BIOSOURCE) to help them optimizing the monitoring and traceability of essential oils. To our knowledge, this is the first time that this type of study was completed in Algeria.

## 2. Results

The selection of species was based on anomalies documented by the company. The objective was to choose oils with high variability, needed by the Algerian market, and derived from wild plants. Based on available data, the following species were chosen: *Thymus capitatus*, *Thymus munbyanus*, *Mentha pulegium*, *Mentha piperita*, *Mentha rotundifulia*, *Rosmarinus officinalis*, *Lavendula stoechas*, *Lavandula aspic*, *Eucalyptus radiata*, *Eucalyptus globulus*, and *Eucalyptus polybractea*.

### 2.1. Chemical Composition of Essential Oils

The preliminary analysis of all extracted essential oils for each species and subspecies revealed qualitative and quantitative differences between the samples from each species. These analyses allowed for cumulating the essential oils of the same species whose chemical profile was qualitatively similar. Table 1 shows class compounds of different collective essential oils. Appendix A describes the detailed chemical composition of each collective essential oil.

The collective essential oil of *Thymus capitatus* from western Algeria featured a single chemotype, with carvacrol as the major compound (69.6%), followed by p-cymene (12.4%) and γ-terminene (4.3%). Appendix A shows a chromatogram of the collective essential oil of *Thymus capitatus*.

The collective essential oil of *Thymus munbyanus* from western Algeria featured three different chemical profiles. Preliminary work revealed the presence of three subspecies of this species with different chemical profiles. The collective oil of the subspecies *abylaeus* was dominated by α-terpinyl acetate (51.7%), α-terpineol (9.7%), borneol (6.8%), and bornyl acetate (5.2%). The collective oil of the subspecies *ciliatus* was dominated by carvacrol (65.7%), followed by γ-terminene (13.6%) and p-cymene (7.9%). The collective oil of the subspecies *coloratus* was dominated by camphor (25.9%), myrcene (16.9%), 1,8-cineole (6.5%), camphene (6.2%), δ-cadinene (5.7%), and limonene (5.6%). Appendix A reports chromatograms of collective essential oils of the *Thymus munbyanus* subspecies.

The collective essential oil of *Mentha pulegium* from western Algeria had a composition rich in pulegone (77.3%), followed by menthone (10.8%). The collective essential oil of *Mentha piperita* was dominated by linalool (40.4%), linalyl acetate (32.6%), and α-terpineol (6.4%). Finally, the collective essential oil of *Mentha rotundifulia* had a chemical profile rich in menthone (28.5%), iso-menthone (19.0%), neo-menthol (10.4%), pulegone (5.6%), and neo-menthyl acetate (5.0%). Appendix A reports chromatograms of the collective essential oils of the *Mentha* species.

The chemical profile of the essential oil of *Rosmarinus officinalis* from eastern Algeria featured two different chemical profiles. One profile had a chemical composition rich in 1,8-cineole (49.4%), camphor (13.2%), α-pinene (6.8%), and β-pinene (5.3%). The other chemical profile was rich in α-pinene (24.4%), camphor (23.5%), camphene (21.5%), 1,8-cineole (9.0%), and β-pinene (6.5%). Appendix A shows a chromatogram of the collective essential oil of *Rosmarinus officinalis*.

The collective essential oil of *Lavandula stoechas* from central Algeria had a chemical profile rich in fenchone (37.8%), camphor (19.5%), and 1,8-cineole (14.8%). The collective essential oil of *Lavandula aspic* was rich in 1,8-cineole (67.7%) and β-pinene (8.0%). Appendix A reports chromatograms of the collective essential oils of the *Lavandula* species.

The collective essential oil of *Eucalyptus radiata* from eastern Algeria was rich in 1,8-cineole (69.9%), α-terpineol (6.0%), limonene (5.7%), and α-pinene (5.4%). The collective essential oil of *Eucalyptus globulus* was rich in 1,8-cineole (80.2%) and α-pinene (6.3%). The collective essential oil of *Eucalyptus polybractea* was rich in 1,8-cineole (42.5%), p-cymene (23.8%), and cryptone (5.9%). Appendix A reports chromatograms of the collective essential oils of the Eucalyptus species.

The chemical profile of the essential oil of *Rosmarinus officinalis* from eastern Algeria featured two different chemical profiles. One profile had a chemical composition rich in 1,8-cineole (49.4%), camphor (13.2%), α-pinene (6.8%), and β-pinene (5.3%). The other chemical profile was rich in α-pinene (24.4%), camphor (23.5%), camphene (21.5%), 1,8-cineole (9.0%), and β-pinene (6.5%).

The collective essential oil of *Lavandula stoechas* from central Algeria had a chemical profile rich in fenchone (37.8%), camphor (19.5%), and 1,8-cineole (14.8%). The collective essential oil of *Lavandula aspic* was rich in 1,8-cineole (67.7%) and β-pinene (8.0%).

The collective essential oil of *Eucalyptus radiata* from eastern Algeria was rich in 1,8-cineole (69.9%), α-terpineol (6.0%), limonene (5.7%), and α-pinene (5.4%). The collective essential oil of *Eucalyptus globulus* was rich in 1,8-cineole (80.2%) and α-pinene (6.3%). The collective essential oil of *Eucalyptus polybractea* was rich in 1,8-cineole (42.5%), p-cymene (23.8%), and cryptone (5.9%).

A total of 122 compounds were identified: 69 compounds in the species of the genus *Thymus*, 71 compounds in the species of the genus *Mentha*, 33 compounds in the genus *Rosmarinus*, 47 compounds in the species of the genus *Lavandula*, and 37 compounds in the species of the genus *Eucalyptus*.

### 2.2. Influence of the Vegetative Cycle on the Yield of Essential Oil

The chemical compositions of, as well as the yields of essential oils from, wild plants vary during the year, as illustrated by the yields of mint essential oil (Figure 1). This figure shows that the yields of essential oil from the same plant vary according to the time and therefore the vegetative cycle of the plant.

Appendix A presents the yields (%) of essential oil extractions expressed in relation to the dry plant.

### 2.3. Variability of Chemical Composition as a Function of Time

The study of the influence of the climatic factor on the production of essential oil reveals that all the compound’s concentrations vary through the seasons. For example, we can point to the chemical composition of the essential oil of *Mentha pulegium*, which shows variations in the percentages of the components of the essential oil. For example, we have selected a sample where the percentage of pulegone is the lowest during the whole year (42.9% at most, contrary to its percentage in the collective oil, which is 77.3%) in order to illustrate the influence of the vegetative cycle on the chemical composition.

Appendix A displays the changing percentages of the dominant compounds in the oil over time. Meanwhile, Figure 2 depicts the chemical variability of *Mentha pulegium*’s essential oil throughout the year. These results indicate that depending on the application of the oil, one can determine the optimal harvest time for the plant and achieve the optimal composition.

The industrial utilization of *Mentha pulegium* essential oil is restricted by its high pulegone content. However, by monitoring the chemical composition throughout the plant’s growth cycle, it is possible to achieve a scenario where pulegone becomes a minor compound in the oil. Specifically, from September to December that pulegone levels decrease, whereas other compounds such as eucalyptol, menthone, and neo-menthol increase. Harvesting the plant during this period offers the advantage of obtaining oil that is more suitable for applications in aromatherapy and the food industry. Additionally, the essential oil yield remains relatively high during this timeframe, ranging between 1.2% and 1.75% of the dry weight of the plant.

By selecting specific harvesting periods, it is possible to obtain an essential oil that is predominantly composed of four different compounds, potentially indicating the presence of distinct chemotypes. These include the pulegone chemotype from July to August, the eucalyptol chemotype from September to November, the menthone chemotype from February to June, and the neo-menthol chemotype from December to January.

This phenomenon is well illustrated by the essential oil of *Mentha piperita*. At the industrial level, this oil is known to exhibit two chemotypes: one rich in linalool and another rich in linalyl acetate. However, through the careful study of the oil’s chemical variability during the plant’s growth cycle, it was discovered that the percentages of dominant compounds vary over time.

Appendix A presents the time-dependent variations in the percentages of the major compounds of this oil.

Figure 3 showcases the annual chemical variability of *Mentha piperita* essential oil. Typically, this oil is recognized for its dominant compound, linalool, followed by linalyl acetate, as observed at industrial level. However, during the period from February to May, there is a convergence for the percentages of these two compounds, with linalyl acetate surpassing linalool as the major compound. It is important to note that this variability arises from the plant’s vegetative cycle rather than the presence of a new chemotype or phylogenetic variation.

### 2.4. Variability of Chemical Composition within a Species

There are two plausible scenarios for the observed variability: environmental or genetic factors. *Rosmarinus officinalis* serves as a notable example in this context. In eastern Algeria, this indigenous species covers a vast area, spanning over 50,000 hectares in the Wilaya of Khenchela, for instance. Chemical analysis of rosemary samples from this region reveals the dominant compound to be 1,8-cineole (49.4%) in the cineole rosemary chemotype, followed by camphor (13.2%), α-pinene (6.8%), and β-pinene (5.3%). However, through multiple samples, it was noted that a second profile emerged, characterized by higher levels of α-pinene (24.4%), camphor (23.5%), camphene (21.5%), 1,8-cineole (9.0%), and β-pinene (6.5%). Although the chemical profiles of both oils are qualitatively identical, they differ quantitatively in the percentages of major compounds. This indicates the presence of two chemotypes, suggesting chemical variability arising from environmental and genetic factors. However, the near-identical profiles of the two oils suggest also that the variability may be driven by environmental factors, particularly soil characteristics. To confirm each hypothesis, it is necessary to genetically identify the different plants and compare the obtained results.

In the industry, rosemary is known to exhibit three chemotypes: 1,8-cineole, camphor, and verbenone. These chemotypes differ not only in their major compounds but also in their overall chemical compositions, indicating genetic factors and phylogenetic variability are involved. However, for rosemary in the Khenchela region, the observed chemical variability between two profiles that have qualitatively similar compositions, is likely to be influenced by environmental factors. Consequently, it can be concluded that there is a single chemotype present in the region, specifically Rosmarinus officinalis CT 1,8-cineole.

*Thymus munbyanus* serves as a second example to highlight the chemical variability within a species. Similar to the aromatic-rich genera, the genus Thymus is known for its polymorphism, which frequently results in the emergence of new varieties or sub-species. In western Algeria, *Thymus munbyanus* is the prevalent species, comprising two coexisting sub-species: *ciliatus* and *coloratus*. These sub-species possess distinct chemical compositions and reside together. The sub-species *ciliatus* is primarily characterized by a high concentration of carvacrol (65.7%), whereas the sub-species *coloratus* is the richest in camphor (25.9%). In industry, users differentiate between the two sub-species using the names *Thymus munbyanus* CT carvacrol and *Thymus munbyanus* CT camphor.

A third subspecies, *abylaeus*, exists within *Thymus munbyanus*, although it is rare and localized. This subspecies displays a unique composition that is notably abundant in α-terpinyl acetate (51.7%), leading to another distinct chemotype for the species known as *Thymus munbyanus* CT α-terpinyl acetate.

Distinguishing the three subspecies of *Thymus munbyanus* visually is very challenging, generating difficulties in plant harvesting, particularly for the common subspecies *ciliatus* and *coloratus*. Using a chemotaxonomic approach for accurate botanical identification in such cases is useful. In conclusion, the variability within *Thymus munbyanus* primarily stems from phylogenetic factors rather than the environmental conditions.

### 2.5. Polymorphism and Chemical Composition

Polymorphism is prevalent in aromatic plant genera, enabling hybridization and the emergence of new species in response to environmental conditions. Specifically, the introduction of Eucalyptus species in Algeria has resulted in a successful acclimatization and diversification of varieties, species, and subspecies.

Our study focused on three commercially important species within the Eucalyptus genus: *Eucalyptus radiata*, *Eucalyptus globulus*, and *Eucalyptus polybractea*. Due to mixed plantings, species differentiation is challenging, resulting in polymorphism and the emergence of regionally acclimated species. Through individual tree sampling, we successfully differentiated between the species using their chemical compositions and aroma profiles. However, distinguishing between the species remains challenging for non-experts, with the differentiating factors residing in small molecular components. Specifically, *E. polybractea* exhibits a distinct odor due to the presence of cryptone. Differentiation between *E. radiata* and *E. globulus* relies on the relative levels of α-terpineol and terpinyl acetate, both are present in both species but with higher proportions in *E. radiata*. In addition, geraniol emerged as a chemical marker, present at an average of 1% in *E. radiata* essential oil, whereas it is absent in *E. globulus* essential oil, enabling the differentiation between these two species based on their unique chemical profiles.

Based on our findings, we can infer that differentiating between the essential oils of Algerian Eucalyptus species has significant challenges due to the propensity of the plants for interbreeding. Accurate identification of each essential oil requires the search for chemical markers that are often present in low percentages. It is important to note that the presence of 1,8-cineole, which is abundant in all Eucalyptus species, does not aid in species identification.

### 2.6. Variability between the Species of a Genus

In general, it is easier to demonstrate chemical variability between two species within the same genus. However, certain exceptions exist. In Algeria, there are two distinct species of wild lavender, *Lavendula stoechas* and *Lavendula aspic*, that often generate confusion. *Lavendula aspic* exhibits high levels of 1,8-cineole (67.7%) in the collective oil analyzed; this is similar to *Eucalyptus globulus*, making it useful for alleviating flu symptoms. Unfortunately, the confusion arises when *Lavandula stoechas*, known for its essential oil rich in fenchone (37.8%) and camphor (19.5%), is erroneously considered toxic. This confusion has detrimental consequences as the essential oil of *Lavandula stoechas* is commonly used externally for its skin-healing properties.

Whereas there is considerable confusion between the *Lavandula stoechas* and *Lavandula aspic* plants, distinguishing their essential oils is less problematic due to their distinct olfactory profiles. *Lavandula stoechas* is characterized by a prominent eucalyptol aroma, whereas *Lavandula aspic* exhibits a more pronounced camphor scent.

Although smell can help in differentiating between the two lavenders, it is not the case for *Thymus capitatus* and *Thymus munbyanus* subsp. *ciliatus*. These thyme species exhibit a striking similarity and share a high content of carvacrol (69.6% for *T. capitatus* and 65.7% for *T. munbyanus* subsp. *ciliatus*). Consequently, differentiation between the two species is challenging, even for experts. Coexisting in the same environment and bearing botanical resemblance, these species also display similar chemical compositions. Accurate differentiation relies on genetic identification or precise descriptions of their oils. Differentiating between the oils of these two species is not possible for non-expert essential oil users.

## 3. Discussion

### 3.1. Essential Oils International Standards

The ISO/TC 54 standard consolidates the norms and standards governing the essential oils marketed by manufacturers. However, these standards may have limitations for newly introduced species, as observed with oils from Algeria such as for example *Mentha piperita*, *Eucalyptus radiata*, and *Lavandula aspic* [46].

In addition, it is crucial to classify newly introduced oils in terms of their chemotypes, major compounds, and the presence of specific marker molecules, even in small percentages. This characterization facilitates the differentiation of oils within the international reference framework. For instance, the essential oil of *Eucalyptus polybractea* is distinguished from other eucalypts by the presence of cryptone, which is considered as a chemical marker despite its low concentration (<5%) [47].

### 3.2. Impact of Chemical Variability on Therapeutic Application

The chemical variability of essential oils presents advantages and disadvantages in their use. In some cases, the chemical profile of an essential oil can restrict its application, as demonstrated by the hepatotoxic compound pulegone that is found as a major component in the essential oil of *Mentha pulegium* [48,49]. The toxicity associated with the major compound, pulegone, limits the industrial use of this oil. However, studying the variation in the chemical composition throughout the vegetative cycle of this species reveals the potential utilization of two compounds highly sought after by the flavor and fragrance industry: eucalyptol and menthol.

The widespread presence of the Thymus genus in Algeria has resulted in its common usage by the population in culinary practices and traditional herbal medicine. The thymol chemotype is valued for its analgesic and anti-inflammatory properties in rheumatology and it is also recognized for its digestive properties, including carminative, aperitive, and cholagogue effects. In contrast, the carvacrol chemotype, which is the predominant compound in the essential oil of *Thymus capitatus*, exhibits similar phenolic properties; however, due to its higher aggressiveness compared with thymol, essential oils containing carvacrol is less commonly utilized [50].

In this context, it is important to understand the concept of chemotype and its influence on the therapeutic application of oils. A chemotype refers to groups of individuals within a species that differ in the presence or absence of one or more chemical substances without there being any macroscopic or microscopic differences between them. The notion of chemotype remains debatable among researchers. Many attribute it to environmental factors (cross-pollination, altitude, microclimate, growing conditions for cultivated species, …), some attribute it to small genetic or epi-genetic differences that have little or no impact on the morphology or anatomy of the species and can thus produce even important changes in the chemical phenotype.

In both cases, this notion leads to considerable changes in the nature of the chemical composition of the essential oils and more particularly the majority compounds, which completely change the vocation of the oil and thus its use. The concept of chemotype is particularly interesting in aromatherapy because different chemotypes can have very different therapeutic qualities, ranging from beneficial to toxic. In the case where this variability is due to environmental factors, we speak of the notion of ecotype.

### 3.3. Parameters Influencing the Chemical Variability of Essential Oils

#### 3.3.1. Mints

External factors such as rainfall, temperature, humidity, life cycle, sunlight, wind, and soil characteristics have the potential to influence the growth and development of plants. Consequently, these factors can lead to substantial variations in the quantity and quality of the produced essential oil, thereby exerting a noteworthy influence on the overall market.

The presence of heavy metals (lead, cadmium, mercury, zinc, nickel, copper, etc.) should be considered due to their potential impact on plant growth and the resulting quality of the extracted essential oil [23].

A decrease in environmental humidity adversely affects plant growth and induces significant biochemical and metabolic alterations, thereby influencing the quality of the essential oil. This phenomenon has been extensively studied in peppermint (*Mentha piperita*), where the impact of external stressors on plant development has been established [51]. Additionally, the unique soil composition, Mediterranean climate, and environmental conditions in Algeria have influenced the chemical composition of Algerian peppermint essential oil, revealing a novel chemotype characterized by high levels of linalool and linalyl acetate. These compounds are highly valued in the fragrance industry for their distinct aromas and find applications in cosmetics, shampoos, soaps, and cleaning products, as well as in the treatment of bruises and as artificial flavors in certain food products [52].

#### 3.3.2. Rosemary

In Algeria, confusion can arise with cineole rosemary essential oil, as it may occasionally contain a significant proportion of camphor, leading to the mistaken belief that it is camphor rosemary. The elevated camphor levels often result from a higher stem-to-leaf ratio or the aging of the plant, which, due to its wild nature, is not regularly pruned. The various chemotypes of rosemary essential oil demonstrate therapeutic potential but require cautious selection to ensure treatment effectiveness and prevent any adverse incidents.

The importance of the rosemary layer in eastern Algeria (more than 50,000 ha of wild rosemary) means that essential oil producers always prefer to harvest wild rosemary instead of planting it. The consequence is that the oil produced is subject to great variations and can sometimes be rejected for export. In order to overcome this problem, we have recommended to the company to work the same plots with a rotation every three years following a good practice guide for the collection of rosemary. This approach will ensure that regular cutting is carried out on the same rosemary plots, which will lead to the rejuvenation of the plants and therefore a greater quantity of leaves compared with the stems, and thus oil closer to the international standard.

#### 3.3.3. Eucalyptus

The problem of polymorphism in the genus Eucalyptus is a serious one in Algeria. The introduction of Eucalyptus species into Algeria was attributed to the *Direction Générale des Forêts* (DGF), which followed the logic of cross planting in order to facilitate the domestication of Eucalyptus species in Algeria and above all the emergence of new hybrids with the capacity to acclimatize better. A few decades later, it has become very difficult to differentiate eucalyptus in Algeria on the basis of the botanical approach and known standards for the classification of eucalyptus. The company BIOSOURCE revealed to us that it was impossible for them to differentiate the Eucalyptus and to select the species radiata because it is the one whose oil is the most profitable. During our investigation, it was also impossible for us to differentiate this species from the others on morphological aspects. Based on the chemotaxonomy and essential oil chemical composition of each tree, we selected *radiata* species from the *globulus* species. We then proceeded to mark each tree in order to facilitate the work of the collectors in the coming seasons. For the *polybractea* species, the problem did not arise because this species could be differentiated from the other two by its characteristic cryptone smell.

#### 3.3.4. Thymus

The case of thymus in Algeria presents two major problems: on the one hand, the essential oils of thymes in Algeria present great chemical variability and, on the other hand, almost all the thyme species are threatened because of illicit cutting and the growing demand of the Algerian market. In order to alleviate these two problems, the only solution is the domestication of wild Algerian thymes and their large-scale planting. In this case, we can promote two qualities of plants: species planted for use in herbal medicine and nutrition and species planted for processing and extracting essential oils. In the second case, it becomes imperative to carry out rigorous varietal selection to produce chemotyped thyme plants that can produce essential oils that meet international standards.

A first experimentation work was launched with the company BIOSOURCE to domesticate and multiply in a nursery a *Thymus capitatus*, giving an essential oil with a carvacrol rate that exceeds 70%, as well as a *Thymus munbyanus* (=*Thymus ciliatus*) giving an essential oil with an α-terpinyl acetate rate that exceeds 55%.

## 4. Materials and Methods

### 4.1. Plants and Extraction of Essential Oils

A total of 323 samples were collected from fourteen selected harvest areas, representing various species and subspecies from five chosen genera. The breakdown of sample locations is as follows: 17 locations for *Thymus capitatus* species; 66 locations for the three subspecies of *Thymus munbyanus* (8 locations for subsp. *abylaeus*, 23 locations for subsp. *ciliatus*, and 35 locations for subsp. *coloratus*); 83 locations for *Mentha rotentifolia* species; 53 locations for *Mentha pulegium* species; 30 locations for *Mentha piperita* species; 31 locations for *Rosmarinus officinalis* species; 15 locations for *Lavendula stoechas* species; 12 locations for *Lavendula aspic* species; 3 locations for *Eucalyptus radiata* species; 10 locations for *Eucalyptus globulus* species; and finally, 3 locations for *Eucalyptus polybractea* species. Detailed information on species stations, localities, voucher codes, GPS coordinates, yields, and altitudes can be found in Appendix A.

The essential oils were obtained through hydrodistillation of fresh aerial parts using a Clevenger-type apparatus, following the guidelines of the *European Pharmacopoeia* [53]. Distillation durations ranged from 3 to 4 h. The distillation yields of essential oils, expressed as a percentage relative to the weight of the dry aerial parts (*v*/*w*), are as follows: 0.50–0.73% for *Thymus capitatus;* 0.84%-1.38% for *Thymus munbyanus* subsp. *munbyanus*; 0.49–0.96% for *Thymus munbyanus* subsp. *eu-ciliatus*; 0.29–0.58% for *Thymus munbyanus* subsp. *coloratus*; 0.41–3.01% for *Mentha rotentifolia*; 0.29–1.75% for *Mentha pulegium*; 0.26–1.33% for *Mentha piperita*; 0.50–1.15% for *Rosmarinus officinalis*; 0.38–0.99% for *Lavendula stoechas*; 0.47–1.07% for *Lavendula aspic*; 0.41–1.78% for *Eucalyptus radiata*; 0.54–2.03% for *Eucalyptus globulus*; and 0.35–1.42% for *Eucalyptus polybractea*.

### 4.2. Identification of Compounds

The identification of individual compounds was conducted through the following methods: (i) comparison of non-polar column retention indices with established values from the literature [54,55]; (ii) comparison of mass spectra with entries in commercial libraries [56]; and (iii) comparison with the retention indices and mass spectra stored in our laboratory’s own database that were acquired under identical operating conditions.

### 4.3. Quantification of Compounds

The quantification of compounds in the essential oils followed the method described in [57], with adaptations made by our research team [9]. The quantification involved normalizing the peaks using response factors obtained from the FID relative to tridecane (0.7 g/100 g), which was used as an internal standard. The results were expressed as normalized percentage abundance.

### 4.4. GC Analysis

Essential oil samples were analyzed using a Thermo Scientific Focus GC equipped with a flame ionization detector (FID). The analysis employed a DB-5 fused-silica capillary column (30 m × 0.25 mm × 0.25 μm) and an SPB-1 with helium as the carrier gas (flow rate of 1.0 mL/min). A 1 μL injection volume of a 2% solution in cyclohexane was used, with an initial pressure of 1.0 Pa. The column temperature was programmed as follows: 60 °C for 3 min, followed by a gradient from 60 to 240 °C (3 °C/min) and from 240 to 300 °C (10 °C/min), with a final hold at 300 °C for 10 min. The injector temperature was set at 250 °C, and the injection mode was “splitless.” The analysis duration was 80 min. Triplicate analyses were conducted for each sample. The retention indices (RIs) of the compounds were determined by comparing their retention times (RT) with those of a series of n-alkanes (C7–C40) using the Van den Dool and Kratz equation.

### 4.5. GC-MS Analysis

The essential oils were subjected to analysis and identification using mass spectrometry in chromatography coupled to a mass selective detector DSQ II, employing the same previously described conditions. The impact energy used was 70 eV. Qualitative analysis of the compounds relied on comparing their spectral masses and relative retention times with those available in the NIST mass spectra database and Kovats RI on the chromatograms. This comparison was performed in both full scan and selected ion monitoring (SIM) modes, utilizing characteristic ions.

## 5. Conclusions

In several countries, including Algeria, there is currently no specific regulatory framework for essential oils, resulting in their widespread availability in various distribution networks. To ensure quality control, it is imperative to mandate the analysis of the chemical composition of essential oils sold on the market. Additionally, laboratories must establish a traceability code for their batches of raw materials.

Through our study, various approaches were developed to investigate the factors influencing the chemical variability of essential oils marketed by the company BIOSOURCE in Algeria. This study facilitated the establishment of fine collection specifications for wild aromatic and medicinal plants, optimizing collection parameters and data. Consequently, the distillation of essential oils resulted in standardized chemical compositions that adhered to the ISO/TC 54 standard.

Expanding this approach to encompass all collected plants, whether wild or cultivated, holds the potential to standardize cultivation, harvesting, and processing parameters. This comprehensive approach aims to extract essential oils that meet the requirements outlined in the ISO/TC 54 standard. In this regard, our ongoing work will soon include a study that describes the parameters associated with cultivated plants whose oils are marketed in Algeria.

## Figures and Tables

**Figure 1 molecules-28-04439-f001:**
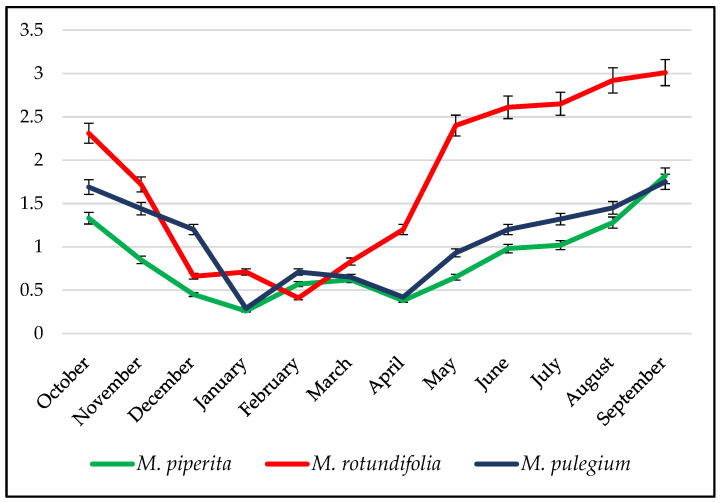
Variation in yields (%) of mint essential oils.

**Figure 2 molecules-28-04439-f002:**
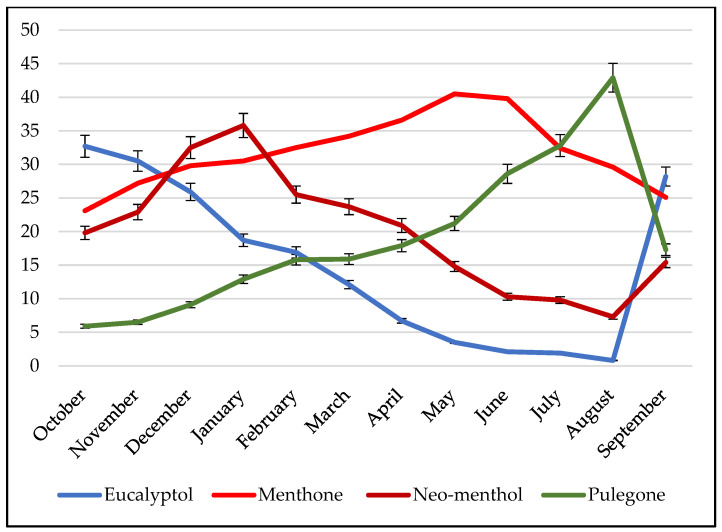
Chemical variability of the major compounds of the essential oil of *Mentha pulegium*.

**Figure 3 molecules-28-04439-f003:**
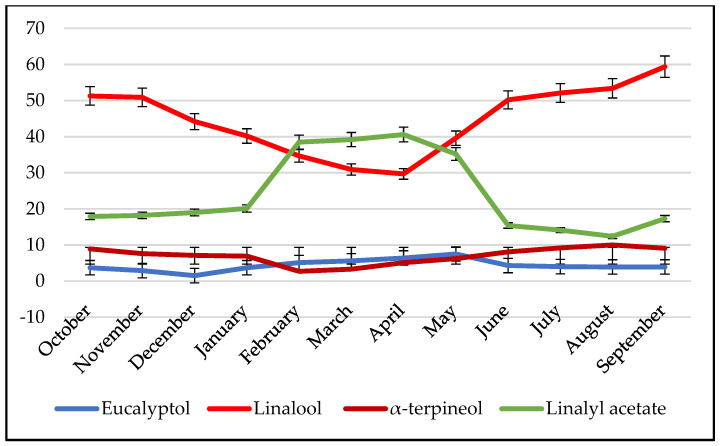
Chemical variability of the major compounds of the essential oil of *Mentha piperita*.

**Table 1 molecules-28-04439-t001:** Chemical composition class compounds of essential oils of different species.

Compound Name and Class	*Thymus*	*Thymus Munbyanus* subsp.			*Mentha*			*Rosmarinus*		*Lavandula*		*Eucalyptus*		
	*Capitatus*	*Abylaeus*	*Ciliatus*	*Coloratus*	*Pulegium*	*Piperita*	*Rotundifulia*	*Officinalis*		*Stoechas*	*Aspic*	*Radiata*	*Globulus*	*Polybractea*
Total Identification %	99.5	99.3	99.7	98.2	98.5	98.8	98.9	93.6	95.3	95.1	95.2	95.7	94.5	95.2
Hydrocarbon compounds	24.7	17	31.3	55.1	2.7	4.8	6.5	21.2	60.9	7.6	17.5	13.7	11.7	41.6
Monoterpene hydrocarbons	23	14.2	28.9	37.9	2	2.8	4.9	20.4	58	6.8	16.4	13.7	11.4	41.6
Sesquiterpene hydrocarbons	1.7	2.8	2.1	17.2	0.7	2	1.6	0.8	2.9	0.8	1.1	trace	0.3	trace
Diterpene hydrocarbons		-	0.3	-										
Oxygenated compounds	74.8	82.3	68.4	43.1	95.8	94	92.4	72.4	34.4	87.5	77.7	82	82.8	53.6
Oxygenated monoterpenes	74.2	81	68.4	40.3	94.2	92.5	91.3	72	34.4	84.9	77.1	81.8	82.8	51.3
Oxygenated sesquiterpenes	0.1	0.5	-	2	0	0.8	0	0.2	0	2.6	0.6	0.2	trace	2.3
Non-terpenic oxygenated compounds	0.5	0.8	-	0.8	1.6	0.7	1.1	0.2	0	0	0	0	0	0

## Data Availability

Data are freely downloadable on the website of BIO-SOURCE company (https://biosource.-dz.com/).

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
