# Peer review of "Chemical Variability and Chemotype Concept of Essential Oils from Algerian Wild Plants"

_molecules, 2023, doi:10.3390/molecules28114439_

Round 1
Reviewer 1 Report
This manuscript addresses an important question about the relationship between secondary metabolites accumulation and growth of plant, which is useful for the gain of maximum yield of target compounds. However, some minor issues still need to be improved: 1.this manuscript needs careful editing in sentences structure to be more concise;2.it would be better to cite more research references in the discussion section, it's more conductive to explain the present results in theory.
Author Response
Thank you for your valuable comments. We have responded for all of them

Reviewer 2 Report
This study identified chemical compositions of eleven wild species of aromatic and medicinal plants native to Algeria, belonging to the genera Thymus, Mentha, Rosmarinus, Lavandula, and Eucalyptu. This manuscript is too wordy and lengthy, data was not well interpreted, and the conclusion was vague and general. So I recommend the manuscript to be rejected.
1. What is soft medicine?Please explain in introduction. See detail in introduction.
2. The introduction is too wordy and lengthy and not well focused. The application of essential oil can be briefly introduced. Please highlight the current research status and problems.
3. The tables should be three-line tables. Please make a revision on tables format.
4. The genera Thymus, Mentha, Rosmarinus, and Lavandula belong to the Lamiaceae family, and Eucalyptus belongs to the Myrtaceae family. Why did you choose these species?Are there any commonalities?
5. Are there criteria for species selection? And are the standards uniform? For example, in the study on the influence of the vegetative cycle on the yield of essential oil, Mentha is taken as the research object. However, there are multiple species of Mentha, why only three species are selected for the study?
6. The table 4 displays of the data may not be directly intuitive. Please change this table into a line graph. In addition, some tables display the same content as pictures. Please keep only the line graphs.
7. In 2.4. the evidence provided for the conclusion that chemical variability of the three chemotypes of rosemary is rather due to genetic factors is not comprehensive. Please provide the of genetic profiles.
8. About Discussion part, it is not sufficiently deep. How do these factors affect chemical variability of essential oils is not discussed.
9. Please add details of sampling collection, including latitude and longitude coordinates, the number of replicated samples.
10. Please polish the language throughout the manuscript.
Author Response
Please find our responses to your comments

Reviewer 3 Report
The context of the work with respect to previous papers is not all clear. References 28-46 describing studies of essential oils in Algeria (the topic of this manuscript) are cited in the Introduction with no explanation. The reader has no clue on what is new about this study and why is it needed. Over 300 samples wee analyzed so this study is quite comprehensive. However, Table 1 is to cumbersome and should be placed in the Supplementary Information. Careful interpretation of Figures 1-3 is not possible because it it not real obvious where the data points are and there are no error bars. Without error bars, one does not know if the changes in the plot trends are significant; perhaps they all tend to overlap. This must be fixed. Triplicate samples were run by GC-FID; was this also the case for GC-MS? There are no chromatograms so it one cannot tell if there might be overlapping peaks that could cause quantitative errors. This must be addressed. There is a lot of Discussion text which should be summarized. The Conclusion section is quite vague, seemingly indicating this is just a quality control study and the phytochemistry is not that important.
Author Response
Thank you for your comments.

Round 2
Reviewer 2 Report
对审查者的答复
This manuscript addresses an important question about the relationship between secondary metabolites accumulation and growth of plant, which is useful for the gain of maximum yield of target compounds. However, some minor issues still need to be improved:
1. The format of the supplementary image Mentha puregium needs to be optimized
2. Have you identified these species? If so, please provide additional appraiser information.
3. In 2.4. the evidence provided for the conclusion that chemical variability of the three chemotypes of rosemary is rather due to genetic factors is still not comprehensive. Differences in precipitation, climate, and altitude can all lead to changes in the content of major chemical components.
4. Supplementary tables 1, 2, and 3 do not indicate the unit of content
5. The discussion is too long. It is recommended to modify the discussion section to specifically discuss the essential oil effects of the 11 species studied, including their pharmacological effects, nutritional effects, and chemical content differences. Research and discussion should be conducted on species of the same genus and different genera, highlighting the differences in essential oil analysis.
6. We suggest supplementing the author's definitions of chemotype concept and chemical variability in the discussion section
7. The article did not mention nutritional effects. It is recommended to modify the title and delete “nutritional effect”
Author Response
Our responses appear in the enclosed document

Reviewer 3 Report
The authors have addressed my concerns except for summarizing Table 1. It is so large I doubt many readers would try to understand it. Certainly all of Table 1 can be in the Supplementary Information and it is described very well in the text. It would be more effective that the manuscript just have the part of Table 1 summarizing the various class of compounds (hydrocarbons and oxygenated compounds) for each plant.
.
Author Response

(The authors gave the same response as above.)
